# HoloLens 1 vs. HoloLens 2: Improvements in the New Model for Orthopedic Oncological Interventions

**DOI:** 10.3390/s22134915

**Published:** 2022-06-29

**Authors:** Alicia Pose-Díez-de-la-Lastra, Rafael Moreta-Martinez, Mónica García-Sevilla, David García-Mato, José Antonio Calvo-Haro, Lydia Mediavilla-Santos, Rubén Pérez-Mañanes, Felix von Haxthausen, Javier Pascau

**Affiliations:** 1Departamento de Bioingeniería e Ingeniería Aeroespacial, Universidad Carlos III de Madrid, 28911 Leganés, Spain; apose@ing.uc3m.es (A.P.-D.-d.-l.-L.); rmoreta@pa.uc3m.es (R.M.-M.); mongarci@pa.uc3m.es (M.G.-S.); dgmato@ing.uc3m.es (D.G.-M.); 2Instituto de Investigación Sanitaria Gregorio Marañón, 28007 Madrid, Spain; calvoharo@yahoo.es (J.A.C.-H.); lydia.mediavilla@salud.madrid.org (L.M.-S.); ruben.perez@salud.madrid.org (R.P.-M.); 3Servicio de Cirugía Ortopédica y Traumatología, Hospital General Universitario Gregorio Marañón, 28007 Madrid, Spain; 4Departamento de Cirugía, Facultad de Medicina, Universidad Complutense de Madrid, 28040 Madrid, Spain; 5Institute for Robotics and Cognitive Systems, University of Lübeck, 23562 Lübeck, Germany; vonhaxthausen@rob.uni-luebeck.de

**Keywords:** augmented reality, Microsoft HoloLens 1, Microsoft HoloLens 2, 3D printing, computer-assisted interventions, orthopedic oncology

## Abstract

This work analyzed the use of Microsoft HoloLens 2 in orthopedic oncological surgeries and compares it to its predecessor (Microsoft HoloLens 1). Specifically, we developed two equivalent applications, one for each device, and evaluated the augmented reality (AR) projection accuracy in an experimental scenario using phantoms based on two patients. We achieved automatic registration between virtual and real worlds using patient-specific surgical guides on each phantom. They contained a small adaptor for a 3D-printed AR marker, the characteristic patterns of which were easily recognized using both Microsoft HoloLens devices. The newest model improved the AR projection accuracy by almost 25%, and both of them yielded an RMSE below 3 mm. After ascertaining the enhancement of the second model in this aspect, we went a step further with Microsoft HoloLens 2 and tested it during the surgical intervention of one of the patients. During this experience, we collected the surgeons’ feedback in terms of comfortability, usability, and ergonomics. Our goal was to estimate whether the improved technical features of the newest model facilitate its implementation in actual surgical scenarios. All of the results point to Microsoft HoloLens 2 being better in all the aspects affecting surgical interventions and support its use in future experiences.

## 1. Introduction

Augmented reality (AR) superimposes computer-generated 3D virtual models to physical objects in real space, enabling users to interact with both worlds simultaneously [1]. This tool has recently been introduced in the clinical field as a more intuitive and natural approach for visualizing complex three-dimensional (3D) data integrated with the actual scenario [1,2] and can improve surgeons’ cognitive skills [3]. Some examples include simulation [4], education [5], clinical care [6], and surgical guidance [1]. So far, surgical navigation has been considered the preferred solution for clinical guidance. However, it uses non-immersive systems which display patient information on external screens. This forces physicians to move their attention away from the patient, disrupting their gaze and reducing intuitiveness (especially among novel users) [7]. Alternatively, AR presents clinical information directly overlayed on the patient [8]. With it, the surgeon can focus on the subject, potentially reducing the surgical time and improving the ergonomics [9].

Mixed reality (MR) is another type of display technology, together with AR and virtual reality (VR). The latter is out of the scope of this work because it completely immerses the users in a virtual environment, making them blind to their surroundings [10]. Hence, it does not add value during surgery. In the other two cases, 3D virtual elements are integrated in the user’s surroundings. The main difference between them is that MR offers the possibility to directly interact with holograms, whereas AR limits the experience to visualization [11]. Nevertheless, many research groups frequently neglect this difference and refer to AR when virtual and real worlds are combined in the visualization [12]. Following their example, we chose the term AR in this work, regardless of the interaction with holograms.

One of the major problems associated with this technology is finding the appropriate transformations to align the displayed objects and the physical world [13,14]. In many cases, this step is performed manually [15,16], but this is not always accurate enough and may require more time and instrumentation [17]. Automatic registration eliminates human error in situations where accuracy is crucial, such as surgical interventions. With that goal, some research groups have used optical [18] or electromagnetic [19] tracking systems to position the AR device in real-world coordinates. Others rely on optical markers and pattern recognition software, such as Vuforia, to achieve this task and improve the hologram stability [20].

AR can be deployed on universal devices, such as smartphones [21] or tablets [22]. However, head-mounted displays (HMDs) are the most frequently chosen option in recent years, as they free the user’s hands and offer a more immersive experience. Microsoft HoloLens is currently considered the best AR platform for surgical practice [14]. It consists of a high-definition, stereoscopic, 3D, optical head-mounted system with a set of grayscale, RGB, and depth cameras which identify the surrounding geometries. Moreover, it includes sensor fusion algorithms which recognize hand gestures and voice commands [23].

According to this, the purpose of our work is threefold. The first is to compare Microsoft HoloLens 1 and Microsoft HoloLens 2 for surgical guidance in orthopedic oncology. We selected these devices because they have already demonstrated their excellence in many clinical applications, including the one we are interested in [1,24,25]. Our main concern was to determine whether the second device improved the AR positioning accuracy. If so, this would promote the implementation of this technology in surgical scenarios. With this purpose, we retrieved the results published in [25]. In that study, we evaluated the Microsoft HoloLens 1 projection accuracy on an experimental scenario using a phantom based on a patient with an extraosseous Ewing sarcoma on a distal leg. The phantom was a modified replica of the patient’s anatomy (namely the tumor and the surrounding bone) and was 3D printed in polylactic acid (PLA). In this work, we developed a similar application for Microsoft HoloLens 2 with comparable features and a comparable interface. With it, we replicated the same experiments and methodology presented in [25] on the same phantom to compare both devices.

Our second goal was to further analyze the Microsoft HoloLens 2 AR projection accuracy. With this purpose, we developed a new experimental scenario based on a patient with an undifferentiated pleomorphic sarcoma on the right shoulder. We 3D printed a larger phantom based on this patient and followed the same procedure as in [25] to analyze the second apparatus in a more complex case. 

The ultimate objective was to test the usability of the newest device in actual surgical scenarios. With this intention, we tested the newest device during the surgical intervention of the second patient and evaluated some of the technical aspects that mainly affected this clinical context: adjustment, intuitiveness, comfortability, and ergonomics. We achieved automatic registration between real and virtual worlds with a 3D-printed, pattern-based marker in all the cases. 

To sum up, this work determined whether Microsoft HoloLens 2 improves the results presented in [25] with Microsoft HoloLens 1 both in projection accuracy and usability for clinical interventions.

## 2. Materials and Methods

Section 2.1 introduces the patients participating in this study. Then, Section 2.2 describes the development of the 3D-printed phantoms and the AR marker, including the sterilization process. In Section 2.3, we describe the AR applications. Finally, Section 2.4 details the system evaluation, including technical features, AR projection accuracy, and surgical experience.

### 2.1. Clinical Data

The data for this study were obtained from two patients treated at the Department of Orthopedic Surgery and Traumatology at Hospital General Universitario Gregorio Marañón (HGUGM) in Madrid. The analysis was performed under the principles of the 1975 Declaration of Helsinki, as revised in 2013 and approved by the Research Ethics Committee at HGUGM, and after obtaining written informed consent from the participants and/or their legal representative. Both patients were assigned an alphanumeric code to preserve their anonymity: HL1HL2_Leg and HL2_Shoulder. Table 1 summarizes the clinical description and CT scan resolution for each patient.

### 2.2. Phantom Development

We developed two 3D patient-specific phantoms from patient CT data, following the steps presented in [21] and summarized in Figure 1. These phantoms (and their dimensions) are presented in Figure 2 and include the patient’s tumor and the surrounding bone structures. Each phantom contained 8 conical holes (Ø 4 mm × 3 mm depth) in the surface for point-based registration. We also designed a surgical guide that fitted onto the patient’s bone to hold a two-dimensional (2D) AR marker (30 × 40 mm^2^). This marker contained a specific pattern in black and white which the AR system could easily recognize. We designed the position of the surgical guide (and the marker within it) in direct coordination with the surgeons considering the surgical approach. By doing so, we could guarantee these elements did not disturb the surgeon’s gaze during the interventions.

The phantoms, surgical guides, and the AR marker for the experimental scenario were 3D printed in PLA with fused deposition modeling (FDM) using a double extruder desktop 3D printer Ultimaker 3 Extended (Ultimaker B.V., Utrecht, The Netherlands). 

To evaluate Microsoft HoloLens 2 during the surgical procedure of patient HL2_Shoulder, we 3D printed a surgical guide in BioMed Clear V1 resin material (Formlabds Inc.) using a stereolithography 3D printer Form 2 (Formlabs Inc., Somerville, MA, USA). This material is USP class IV certified and can be in direct contact with the patient organs for long periods of time [26]. The surgical guide and AR marker were sterilized with ethylene oxide (EtO) at 55 °C [21,27] and 37 °C, respectively. The lower temperature avoided PLA deformation [28]. 

### 2.3. Augmented Reality Applications

We developed the Microsoft HoloLens 1 AR application on Unity 2017.4 LTS (as presented in [25]) and used Unity 2019.10f1 for Microsoft HoloLens 2. The Unity version used was in accordance with Microsoft official recommendations at the time of the experiments [29]. In both cases, we used the C# programming language and Unity Mixed Reality Toolkit (MRTK) (https://github.com/Microsoft/MixedRealityToolkit-Unity, accessed on 28 February 2022) library. 

Even though Microsoft HoloLens incorporates an internal system to track the environment via spatial mapping, this feature could not be used in this study for two main reasons. First, the tracking accuracy achieved with this system ranges from several millimeters to more than one centimeter [30]. This range is not acceptable in surgical interventions, in which a shift of a few millimeters could be the difference between success or failure. On the other hand, it requires identifiable surfaces to work properly. During surgeries, the surface of a patient frequently varies due to body movements, soft tissue displacements, and interaction with surgical tools. Hence, this environment it is not reliable for surface tracking. As an alternative, we used Vuforia SDK (Parametric Technology Corporation Inc., Boston, MA, USA) to track a pattern-based AR marker in the camera field of view (FOV). This marker can be easily attached to a surgical guide, which fits in a unique region of the patient’s bone. This configuration remains fixed during the whole intervention despite the patient’s movements, which allows for the compensation of movements, thus improving the tracking reliability. With it, one can display virtual 3D models in the real environment at specific positions.

The virtual projected models were the anatomical structures of the patients (tumor and surrounding bone), the surgical guides, and the AR marker silhouette [24]. To measure the AR projection error in the experimental scenario, we additionally augmented some control spheres on top of the phantom’s surface. We developed a 3D Slicer module [31] to compute the relative positions between the models and the AR marker. Then, we intrinsically associated these transformations with the virtual models, so that they were always projected in the correct spatial location relative to the marker. By doing so, and considering that the surgical guide (and therefore, the marker) was fixed to the patient’s bone during surgery, we could automatically register the real and the virtual anatomy. In addition, both Microsoft HoloLens’ apps contain an interface with several buttons and sliders to modify the visibility and transparency of every model according to the user’s needs. The user could visually check the registration success by looking at the AR marker silhouette overlaid on the corresponding real object.

### 2.4. System Performance

#### 2.4.1. AR Projection Error

We measured the AR projection error as described in [21]. First, we fitted the surgical guide on the phantom bone, attaching the AR marker for automatic registration. Then, we used a modified version of both Microsoft HoloLens apps to augment fifteen spheres (Ø 3 mm) on the surface of the phantoms (see Figure 3). Three surgical navigation researchers measured this error by recording the position of all the AR spheres with a pointer containing four retro-reflective spheres. We used a Polaris Spectra (Northern Digital Inc. Waterloo, ON, Canada) optical tracking system (OTS) connected to 3D Slicer via PLUS [32] as a gold standard of the measurements. This system could track the spheres of the pointer and we could extrapolate this information to determine the actual position of the pointer’s tip. Later, we compared the location of each sphere in the real world with their expected positions to measure the deviation in the projection of the holograms (AR projection accuracy) [21,25]. 

Each user performed this process three times, removing and replacing the AR marker between repetitions. We decided to use the same number of users to perform this experiment as in [25], as well as the same number of repetitions, to make the Microsoft HoloLens 2 results comparable to the ones obtained with Microsoft HoloLens 1. We calculated the root mean square error (RMSE) and median error from the recorded data, as described in [1]. The origin of coordinates for all the measurements was the center of the AR marker pattern surface (recall Figure 3). The Z axis was defined as the vector normal to that surface. The transversal and longitudinal axes of the marker corresponded to the X and Y axes, respectively. The position of the HL1HL2_Leg phantom spheres was measured with Microsoft HoloLens 1 (data published in [25]) and Microsoft HoloLens 2. The evaluation in phantom HL2_Shoulder was only performed with Microsoft HoloLens 2. We performed some statistical tests to analyze the differences among the distinct classifications of the data.

#### 2.4.2. Technical Differences and Survey

We compared the technical aspects of Microsoft HoloLens 1 and Microsoft HoloLens 2 which mainly affected our specific surgical application (such as weight, hologram interaction intuitiveness, or comfortability) and evaluated them for orthopedic oncological surgeries. To perform this evaluation, we created a survey in which users who had tried both models in this clinical context had to rate these features on a scale from 1 (terrible) to 5 (perfect). We based our survey on other studies which created similar questionnaires to evaluate their systems, such as [21,33,34]. This questionnaire can be found in the Appendix A and was answered by three orthopedic oncological surgeons and four surgical navigation researchers. We analyzed the results considering the Microsoft HoloLens 1 [35] and Microsoft HoloLens 2 [23] official datasheets.

#### 2.4.3. Surgical Experience

In addition to evaluating the AR projection accuracy, we considered measuring the usability of Microsoft HoloLens 2 in actual surgical scenarios to be of great importance. Our goal was to analyze the glasses in terms of comfortability and ergonomics, considering the long duration of these surgical interventions. For this reason, we tested the Microsoft HoloLens 2 app during patient HL2_Shoulder’s surgery and evaluated the overall experience. The orthopedic surgeon put on the Microsoft HoloLens 2 before starting the intervention and raised the visor. He first performed an incision over the humerus until the bone was reached. Then, he fitted the surgical guide, fixed it with screws, and introduced the AR marker into the surgical guide socket. After that, a nurse lowered the visor on the glasses to start the augmented reality visualization. 

The virtual models of the tumor and the bone were automatically registered with the actual anatomical structures of the patient. The surgeon used the app buttons and sliders to tailor the visibility of each augmented model and assess the alignment of the projections. The surgery followed the pre-established surgical plan, and the tumor was completely resected. The surgical procedure continued without incidences.

## 3. Results

### 3.1. AR Projection Error

The Vuforia SDK detected the AR marker almost immediately in both applications. In addition, we could visually confirm a correct registration between the real and virtual world based on the good alignment of the AR marker silhouette with the actual AR marker. Figure 4 displays the AR projection error measured in phantom HL1HL2_Leg with Microsoft HoloLens 1 (Figure 4a) and Microsoft HoloLens 2 (Figure 4b) and in phantom HL2_Shoulder with Microsoft HoloLens 2 (Figure 4c), grouping the results by user. The column “All Users” combines the results obtained by the three individuals in a single box. A Kruskal–Wallis statistical test proved no significant differences between users.

In addition, Table 2 shows the AR projection RMSE and median error for each device and patient. These numbers reveal that the AR projection error in patient HL1HL2_Leg was lower with Microsoft HoloLens 2 (RMSE = 2.165 mm and median error = 1.734 ± 1.224 mm) than with Microsoft HoloLens 1 (RMSE = 2.833 mm and median error = 2.587 ± 1.501 mm). These differences between devices were significant on a Mann–Whitney U-test *(p*-value < 0.05). 

To understand the discrepancy between the error obtained with Microsoft HoloLens 2 for phantoms HL1HL2_Leg and HL2_Shoulder (RMSE 2.165 mm and 3.108 mm, respectively), Table 3 shows the median distance of the AR control points to the origin of coordinates (center of the AR marker). Control points were distributed all over the phantom’s surface, and phantom HL2_Shoulder was much larger (recall Figure 2). On average, control points were 3 cm further from the origin than in phantom HL1HL1_Leg. A Mann–Whitney U-test confirmed that the control point distribution of both phantoms was significantly different. As a result, we cannot compare their AR projection errors. To complement this table, Table 4 organizes the median error and the IQR of all the experiments by axis.

Finally, Figure 5 shows the AR projection error in each individual control point for each phantom and device. We performed a post hoc Connover test on data from phantom HL1HL2_Leg and found significant differences in point 7, 13, 14, and 15 with respect to the rest of the points. These four are, in fact, the points with the largest error in Figure 5a. This pattern was repeated in phantom HL2_Shoulder. In this case, the measurements acquired over the tumor were significantly different from those over the bone, except for point 11 (which was significantly different from the other tumor points). For this analysis, we highlighted the points that had significant differences with at least three other points (see Appendix A). 

### 3.2. Technical Differences and Survey

Figure 6 is a simplified representation of some survey responses comparing Microsoft HoloLens 1 and Microsoft HoloLens 2. The average score was 2.84 out of 5 for Microsoft HoloLens 1 and 4.00 for Microsoft HoloLens 2. In other words, Microsoft HoloLens 1 was evaluated as bad–neutral and Microsoft HoloLens 2 as good.

### 3.3. Surgical Experience

A video included in Appendix A shows how the pattern recognition was almost immediate and not disrupted by blood or illumination conditions. Upon detection, all the AR models (including bone structures, tumor, surgical guide, and AR marker contour) were automatically displayed in place (see Figure 7), allowing a better evaluation of the tumor limits. The slight projection shift visible in the figure was caused by the misalignment between the Microsoft HoloLens 2 recording camera and the user’s view. Consequently, the captured images did not correspond exactly with the user experience, who saw the models correctly aligned.

The surgeon gently moved the patient’s arm to guarantee that the AR marker remained fixed in the bone, checking how the virtual models followed the movement to always align with the anatomical structures. In addition, interaction on the user interface was intuitive, effective, and allowed the surgeon to accommodate the holograms’ visibility to his needs. When no longer needed, the AR marker was easily detached, and a nurse raised the Microsoft HoloLens 2 visor again to continue with the intervention. Indeed, the surgeon highly appreciated this feature because it greatly improved ergonomics. In addition, he acknowledged the lightness and comfortability of the new headset compared with its predecessor.

## 4. Discussion

In this work, we analyzed Microsoft HoloLens 2 improvements with respect to Microsoft HoloLens 1 in orthopedic oncology. We firstly assessed the AR projection accuracy of both devices and then compared some technical aspects, including comfortability and intuitiveness. To do so, we retrieved the data published in [25] which analyzed AR projection accuracy with Microsoft HoloLens 1 over a 3D-printed phantom. This phantom was based on a patient treated at Hospital General Universitario Gregorio Marañón. Then, we developed a new app for Microsoft HoloLens 2 to acquire new data on the same phantom. Comparing the results obtained from both devices, we could estimate the AR projection improvements from one device to the next. To further test the capabilities of Microsoft HoloLens 2, we developed a new phantom based on a second patient from the same hospital and measured AR projection accuracy with Microsoft HoloLens 2 following the same protocol. To finish, we tested our Microsoft HoloLens 2 app during the surgical intervention of this patient. In both cases, the apps projected 3D virtual models created from patients’ anatomy overlayed on the actual patients (or the corresponding phantom in the experimental scenario). 

Currently, most studies that apply augmented reality in medicine perform a manual registration between real and virtual worlds [8]. Although easy to execute, the resulting system accuracy is not acceptable for surgical applications. Our proposal achieves automatic registration by utilizing patient-specific surgical guides that fit in a specific region of the patient’s bone. The procedure followed in this study is a step forward from the one presented in [21], where we showed the AR information on a smartphone.

In terms of the results regarding the AR projection error, the RMSE measured with Microsoft HoloLens 2 on phantom HL1HL2_Leg was almost 0.7 mm lower than with Microsoft HoloLens 1, which translated to a more accurate AR projection. The second experiment with Microsoft HoloLens 2 (on phantom HL2_Shoulder) resulted in an RMSE 1 mm larger than in the first case. This increased error is explained by the larger size and shape of the second phantom. With respect to the errors by axis, phantom HL1HL2_Leg yielded much lower error in the Z axis (normal to the AR marker surface) than in the other two axes. This can be attributed to the short depth of this mannequin compared to the other dimensions. Concerning phantom HL2_Shoulder, there were no meaningful error differences in each axis, as the phantom size was more similar along the three dimensions. 

Concerning the error in each control point, we explored possible correlations between AR projection error and distance to the origin (Appendix A). However, we found no evidence of larger errors in further distances. This suggests that AR projection is robust in these working volumes [36]. In turn, statistical tests on phantom HL1HL2_Leg demonstrated a significant difference between the points lying on a more irregular surface (e.g., the tumor) and points on the planar bone surface. In phantom HL2_Shoulder, there were also significant differences between easy- and difficult-to-access points. This could be the result of having a planar AR marker, since the users could not freely move around the phantom, but they had to keep their head more or less still in front of the AR marker surface, so the Microsoft HoloLens did not lose tracking. For this reason, those points that would require a different perspective showed the largest error. Nevertheless, neither the relation between the error at each point nor the orientation of the surface on which this point lay was linear. Our results point out to a more complex combination of the angle, perspective, and comfortability of the user acquiring the data.

Even so, the errors obtained in this study were much lower than the ones presented in [1], which proposed a new method for orthopedic surgical guidance with Microsoft HoloLens. They also measured system error by recording the position of some randomly distributed fiducials over the surface of a phantom, using an optical tracking system as the gold standard. Like us, they performed an automatic registration between the real and the virtual structures but required an ultrasound probe to complete the alignment by locating bony structures that were then registered to the CT. With that approach, they reported an RMSE of 36.90 mm and component errors in the x, y, and z directions of 4.10, 3.02, and 43.18 mm, respectively. They also attribute the large Z component error to Microsoft HoloLens depth perception limitations. In our case, we obtained a maximum RMSE of 3.018 mm and a maximum error per axis of less than 1.8 mm in Microsoft HoloLens 1 and less than 1.4 mm in Microsoft HoloLens 2. In addition, our registration method did not imply any costly or time-consuming instrumentation and reduced the RMSE more than ten times. 

With respect to the capabilities and technical differences, the newest device displayed great benefits compared to the first one. These results were expected considering the Microsoft HoloLens 2 technical datasheet and new features. First, even though the difference in weight is not very high (579 g in the first model versus 566 g in the second one), Microsoft HoloLens 2 relocates the battery pack on the back of the head, improving weight balance. In addition, it incorporates a dial-in fit system and a flipping visor. These highly increase the ergonomics and make the headset more comfortable for extended sessions. These two aspects were the most appreciated by the surgeon during the intervention of patient HL2_Shoulder. 

Regarding the visualization, the diagonal field of view of the second device (52°) highly surpasses the first one (30°). This larger value enhances the integration of the holograms in the 3D space and offers a more immersive experience. On the other hand, Microsoft HoloLens 2 greatly improves hand-gesture recognition. While Microsoft HoloLens 1 interaction was simply oriented to use the head like a mouse pointer and the finger tap gesture as the mouse button, the newer device recognizes twenty-five articulation points per hand considering wrist and fingers. This aspect permits a more intuitive interaction to grab, translate, rotate, and scale holograms. It was especially beneficial during surgical intervention for patient HL2_Shoulder, in which the surgeon could interact with the buttons’ interface as in an actual panel. Finally, Microsoft HoloLens 2 has upgraded processing power and RAM, providing a faster response speed and a more detailed experience.

Once again, augmented reality has proven to be very promising in these surgical applications. AR head-mounted displays show relevant information in front of the surgeons, freeing their hands. In addition, new incorporations on Microsoft HoloLens 2 enhance device ergonomics and accuracy. Hence, it does not seem unreasonable to expect the introduction of these devices in clinical practice in the near future. Nevertheless, several aspects still have to be improved. On the one hand, Microsoft HoloLens sterilization is still not possible, so surgeons cannot touch the glasses at any point during the intervention. In addition, our solution relies on solid structures (i.e., bones) to hold the surgical guides, so it cannot yet be directly extrapolated to surgeries only involving soft tissues. To finish, the AR marker detection is continuous, so the marker must be visible at all times, which might be inconvenient at some points. To overcome this limitation, we assigned different colors to the augmented AR marker silhouette according to the tracking state. These colors were green when the AR marker was visible and red when the tracking was lost. In the latter case, the virtual models remained in the last place the tracker was detected. Hence, the surgeon could always see the models, but he had a warning to see the reliability of the projection in real time. All of this was reflected in the surgeon’s positive feedback during the surgical experience.

## 5. Conclusions

This work analyzed the improvements in Microsoft HoloLens 2 over Microsoft HoloLens 1 in terms of AR projection accuracy. To do so, we developed an AR application for Microsoft HoloLens 2 based on the one presented in [25] and used the same 3D-printed, patient-specific phantom and methodology to compare the AR projection capabilities of both devices in an experimental scenario. The results show an enhancement of almost 25% from the first to the second model. 

Then, we developed a larger patient-specific, 3D-printed phantom to further analyze Microsoft HoloLens 2’s AR projection accuracy. In addition, we tested the newest device during the surgical intervention of the second patient. During this, we evaluated the technical aspects that mainly affect this clinical application. These include adjustment, hologram quality and interaction, and hand recognition. 

All of the procedures were based on patient-specific surgical guides and an AR marker containing easy-to-identify patterns which was detected using Vuforia SDK. To our knowledge, this is the best option for automatic registration and clearly recommends Microsoft HoloLens 2 rather than Microsoft HoloLens 1 in clinical interventions. We hope this work helps others decide which device to use in similar applications and that it establishes the basis for future studies of augmented reality in this clinical context.

## Figures and Tables

**Figure 1 sensors-22-04915-f001:**
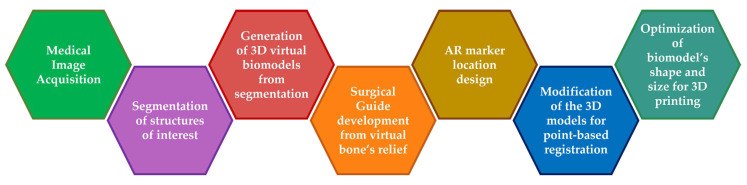
Diagram of the workflow presented in [21] to develop a patient phantom and a surgical guide.

**Figure 2 sensors-22-04915-f002:**
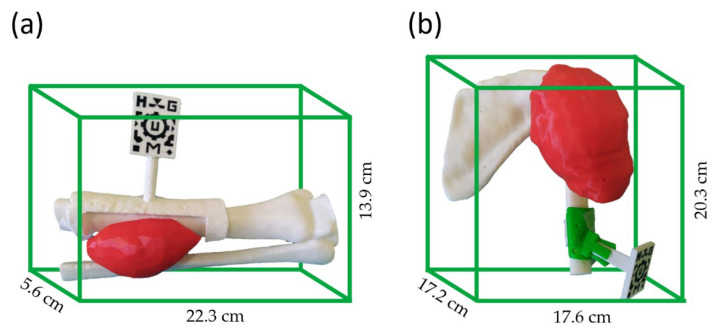
The 3D-printed phantoms based on patient (**a**) HL1HL2_Leg and (**b**) HL2_Shoulder, with dimensions. Bone structures were 3D printed in white and tumors in red. Surgical guides were fitted onto the bone surface and held the AR marker.

**Figure 3 sensors-22-04915-f003:**
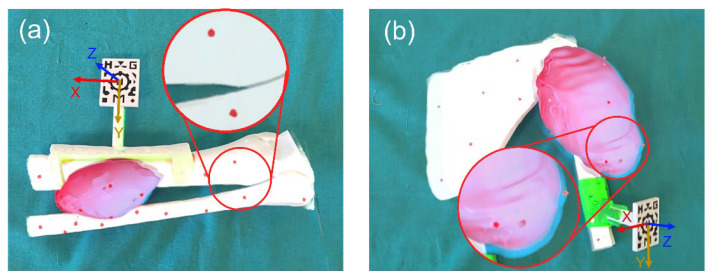
Control spheres projected over (**a**) phantom HL1HL2_Leg and (**b**) phantom HL2_Shoulder with Microsoft HoloLens 2.

**Figure 4 sensors-22-04915-f004:**
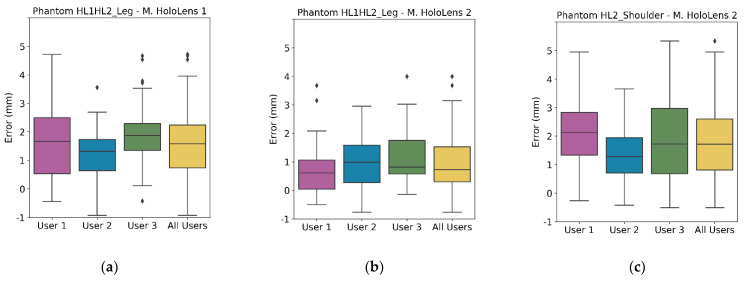
AR projection error for (**a**) Microsoft HoloLens 1 and (**b**) Microsoft HoloLens 2 in phantom HL1HL2_Leg, and (**c**) phantom HL2_Shoulder, grouped by user. Each box includes from first to third quartile of the dataset, with middle line indicating the median, and whiskers for the highest and lowest values (±1.5 times the standard deviation). Black diamonds represent outliers.

**Figure 5 sensors-22-04915-f005:**
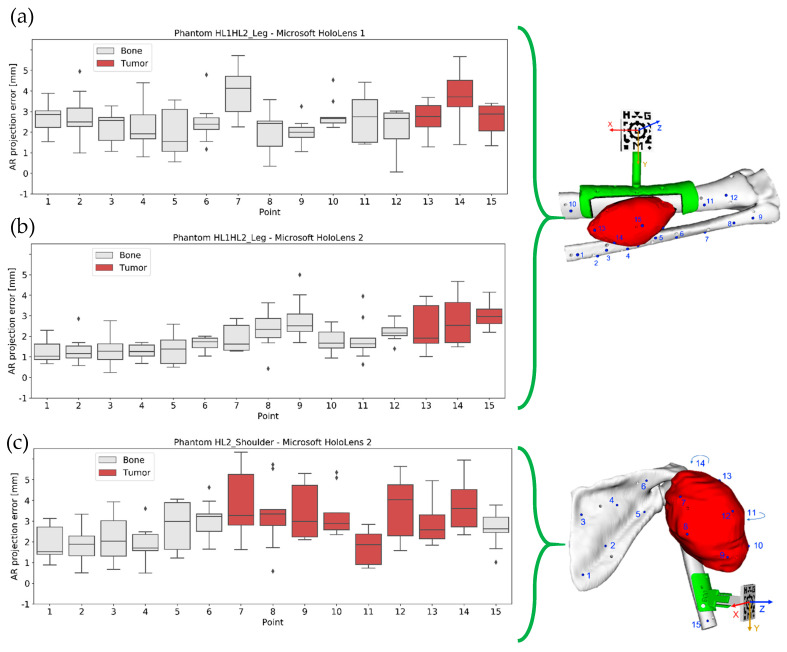
AR projection error by point obtained for patient (**a**) HL1HL2_Leg with Microsoft HoloLens 1, (**b**) HL1HL2_Leg with Microsoft HoloLens 2, and (**c**) HL2_Shoulder with Microsoft HoloLens 2. On the right, positions on the virtual control points over the phantoms. Black diamonds represent outliers.

**Figure 6 sensors-22-04915-f006:**
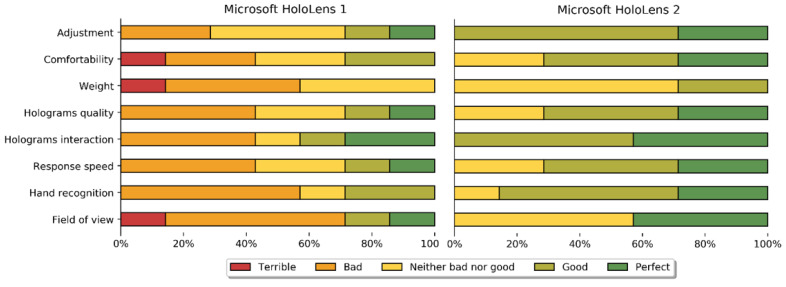
Survey responses regarding Microsoft HoloLens 1 and Microsoft HoloLens 2.

**Figure 7 sensors-22-04915-f007:**
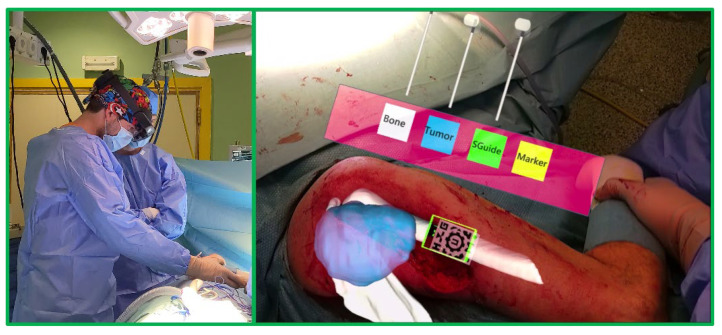
AR visualization on Microsoft HoloLens 2 from surgeon’s perspective during the intervention on patient HL2_Shoulder. The left-hand side of the figure shows an external view of the surgical field.

**Table 1 sensors-22-04915-t001:** Clinical description and CT scan resolution for each patient involved in the study.

Patient ID	Gender/Age	Diagnosis	Tumor Location	CT Resolution (mm)
HL1HL2_Leg	F/17	Extraosseous Ewing’s sarcoma	Distal leg	0.98 × 0.98 × 2.50
HL2_Shoulder	M/50	Undifferentiated pleomorphic sarcoma	Shoulder	1.22 × 1.22 × 3.00

**Table 2 sensors-22-04915-t002:** AR projection errors divided by device and phantom.

Device	Patient ID	Root Mean Square Error (RMSE) (mm)	Median/Interquartile Error (IQR) (mm)
Microsoft HoloLens 1	HL1HL2_Leg	2.833	2.587/1.501
Microsoft HoloLens 2		2.165	1.734/1.224
	HL2_Shoulder	3.108	2.717/1.787

**Table 3 sensors-22-04915-t003:** Median distance of the AR control points to the origin of coordinates for each axis.

Phantom ID	Median Distance to Origin/IQR [mm]	Median Distance to Origin in X Axis/IQR [mm]	Median Distance to Origin in Y Axis/IQR [mm]	Median Distance to Origin in Z Axis/IQR [mm]
HL1HL2_Leg	108.011/19.667	39.537/42.119	96.310/19.872	11.901/8.888
HL2_Shoulder	144.308/84.185	47.505/60.248	96.944/49.271	49.430/102.559

**Table 4 sensors-22-04915-t004:** Median error of AR control points per phantom, device, and axis.

Device	Phantom ID	Median Error/IQR in X-Axis	Median Error/IQR in Y-Axis	Median Error/IQR in Z-Axis
Microsoft HoloLens 1	HL1HL2_Leg	1.713/1.611	1.187/1.345	0.587/0.952
Microsoft HoloLens 2		1.172/1.054	0.574/1.056	0.643/0.719
	HL2_Shoulder	1.343/1.525	1.168/1.120	1.061/1.522

## Data Availability

Data are contained within the article or Appendix A.

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
