# Peer review of "HoloLens 1 vs. HoloLens 2: Improvements in the New Model for Orthopedic Oncological Interventions"

_sensors, 2022, doi:10.3390/s22134915_

Round 1

Reviewer 1 Report

Authors showed some improvements of the HoloLens 1 and 2 for clinicl approach. However, there are some limited data for the analysis. Therefore, authors need to revise the manuscript.
Literature background for this manuscript is also limited.

1. In Figure 4, there are Users 1,2,3 and all users. How many users shown in this manuscript ?
2. Authors showed some projection error in Figure 5. Are there any necessary more analysis for projection errors ? For the manuscript, authors must provide some other parameter analysis such as angle differences.
3.In Tables 3 and 4, authors had better combine these 2 into one Table if needed.
4. How authors select Holo lens 1 and 2 ?
5. In Table 3, there are only some analysis for x, y, and z axes. Are these 3 axes analysis suitable for AR control points ?
6. Authors need to provide some previous research similar to this article in the introduction section.

Author Response

Dear Reviewer,

We want to thank you for your insightful comments, questions, and your interest in our work. The replies to your reviews are in the attached file.

Thank you for your time reading and analyzing our paper, as well as for your kind comments and suggestions. We believe that the proposed changes are valuable and improve the quality of the document.

Reviewer 2 Report

In the manuscript entitled “HoloLens 1 vs. HoloLens 2: Improvements of the new model for clinical interventions.”, the authors propose to analyzes the Microsoft HoloLens 2 utilization in this context and compares it to Microsoft HoLoens 1. The results show that almost 25% improvement can be obtain compared to the Microsoft Holoens 1, which greatly enhance the comfortability, usability and ergonomics for surgeons.. Firstly, they used a 3d printed phantom based on a patient with an extraosseous Ewing sarcoma on a distal leg. Then, the comparison was conducted between the second model and the first one via the new developed app. Finally, they tested HoloLens 2 during the actual surgery of the patient, which showed the clinical application of this surgical guidance approach. The concept and demonstrations were also introduced to show its novelty and superiority. Considering this work is of great scientific significance and practical application potential, I would like to recommend the publication of this manuscript in Sensors after the following issues are addressed.

1, It’s very interesting to see how the new app were used, details including processing method, versions (it is different for the old version?), should presented in the manuscript.

2, The English language construction of the manuscript is very clumsy and it needs to be carefully checked and polished to avoid typos.

3, Full names of all proper nouns should be indicated when they first appear in the text.

4, Some Figures were recommended to merge as one, this would make it more like a research article than experiment report.

5, More abundant structural characterization methods should be used to verify the effects and differences of different printing material formulations in practical applications, and it is also more intuitive and attractive to readers.

6, Some up to date literatures are recommend to cite: Engineered Regeneration 2 (2021) 175-181 (https://doi.org/10.1016/j.engreg.2021.12.002)

Author Response

Dear Reviewer,

We want to thank you for your insightful comments, questions, and your interest in our work. The replies to your reviews are in the attached file.

We are certain that your suggestions will help to increase the value of the manuscript.

Reviewer 3 Report

The manuscript presents the development and application of marker-based guides for surgery guidance using the Microsoft HoloLens. An error comparison is performed between the Microsoft HoloLens 1 and HoloLens 2 with a patient with an undifferentiated pleomorphic sarcoma on the right shoulder.

The topic is relevant given the mass adoption of immersive technologies in the health care field. The paper excels when discussing the use case, the approach to reduce error in tracking, and how it was used on a live procedure. However, the paper falls short in many other areas that require improvements. For example, the chosen narrative over the improvements of the HoloLens 2 over the HoloLens 1 needs to be better addressed as these are evident by reading the differences in the technical features. Here, the discussion around improving the tracking within the context of the use case is more interesting. The study is not presented, a statistical analysis is conducted, but no information on samples or the decision-making is clear.  At some point, it is indicated that there were three users, but with the procedure, it seems to have been just one. Because of this, it is difficult to agree on the generalization of the results. The survey used to measure perceptions between the HoloLens devices seems to have been used with a scenario other than the one created, this needs to be clarified. Also, why wasn't a standardized questionnaire used? Because of the previous comments, the methods and results need to be better covered. After reading the manuscript, it feels like the contribution is not the comparison between the two headsets, but the improvement in tracking by using guides. However, this is not explained in-depth and markerless solutions should be discussed within the scope of the work to better understand the choice of markers. The discussion suffers from a focus on the technical features of the headsets, and just the last paragraph covers aspects relevant to the study. Finally, the conclusions do not meet the expectations and lack support from the data provided.

Please see the PDF for additional comments and suggestions, and the bullet points below:

Typically, the HoloLens is associated with and advertised as a Mixed Reality device, because of this, it is important to acknowledge and address the differences between MR and AR and why AR has been chosen to be used with the HoloLens.

What does it mean "we obtained an improvement of..." What did you develop on top of the technology for achieving this result? This is confusing as the hardware for the Hololens 2 presents signigicant improvements that contributed to this.

The abstract does not present the research question or gap that the paper addresses. The outcome is somewhat expected given the improvements made to the hololens 2.

Consider explaining why AR has been adopted when most work associated with the HoloLens is mixed reality. AR is not that recent, it has become popular because of technological advances. What do recent mean in the context of the manuscript?

Please explain what makes AR intuitive and natural. Traditionally, AR requires the user to hold a device and look at the world through a small screen.

Why is AR alignment a major problem? Current AR solutions employ a combination of technicals that no longer require markers because these are able to extract information from the environment. This is something the HoloLens does. This paragraph could be stronger if it focuses on the limitations of the HoloLens tracking system. 

The goal of the papers to compare intuitiveness, comfortability, and projection accuracy needs to be consistent with the previous statement. Based on the tech features of the devices, it can be anticipated that the HoloLens 2 will outperform its predecessor, which begs to question of why is this a goal? Is the accuracy correlated to the previous parameters?

The end of the introduction discusses the tests. However, the research question and rationale is unclear. While it is interesting and valuable to have used the HoloLens 2 in a real setting, more details are needed to understand the scientific choices that led to this. It would seem that this test could demonstrate transferable skills, but the lack of information makes it difficult to understand. 

Due to how the HoloLens projection technology works, why do you indicate that it is an AR projection? Microsoft refers to it as a holographic projection. https://docs.microsoft.com/en-us/windows/mixed-reality/discover/hologram 

In Section 2.1 each HoloLens seems to have been tested with a different patient's model. Why was this chosen instead of having both devices tested with both scenarios? 

Section 2.2 introduces the phantom devices. However, it is not indicated why the name phantom was chosen. Replicas seem more appropriate. Please summarize the steps from [18] as otherwise, readers will have to stop, browse the citation, and read that source, thus breaking the flow of reading. 

It is stated that the markers were chosen because they are easy to recognize. This choice does not take advantage of the HoloLens tracking features that enable markerless tracking. Using markers restricts the view from which these are visible. Including a rationale for this will strengthen the manuscript. I believe that the surgical guides could help with this comment.

Using subfigures a and b with their own captions will help better provide details for both figures. 5.6 cm and 17.2 cm measures are not clear, what are they representing?

Why are there two different versions of the Unity game engine being used?

Ensure that all HoloLens references are properly written, most of the document refers to HoloLens instead of the Microsoft HoloLens.

When comparing the technical features of both HoloLens devices, why was the focus weight, hologram interaction intuitiveness, or comfortability? Why weren't the field of view, projection distance, tracking, camera resolution, spatial audio, and other features compared? In Section 2.4.1 seems that a usability evaluation is being conducted, why not use a standardized questionnaire such as the System Usability Scale?

Figure 2 could use some labels to better understand tracking error.

When describing the tracking error, it is indicated that three users performed the process three times. Why three users? Who were these three users? What is the background of these three users? Why three times? How does this study design ensure that an appropriate number of samples were collected to make these finding generalizable? This will strengthen the stats performed.

There seems to be missing information, Figure 3 shows results comparing both HoloLens in terms of adjustment, comfort, weight, hologram quality (not AR which is confusing), hologram interaction, response, hand recognition, and field of view. However, no study design within the context of the use case was presented to support these responses. It seems as it the survey focused on out-of-the-box features and not the use case, which is problematic. These responses should have been associated with the users employing the HoloLens devices in both use cases.

It is indicated that the Vuforia SDK perfectly detected the AR marker. What does perfectly mean? This is not only because of Vuforia, what about the Hardware and the image processing performed by the HoloLens?

If the tracking was perfect, Figure 4 needs to be better explained. 

Figure 6 is difficult to read because of its lower quality. 

The discussion is summarizing what was done but is not discussing the results. The second paragraph seems more adequate for a conclusion. The comments on how better the HoloLens 2 perform are not novel and expected, I was expecting more discussion on how the users performed with your tool with respect to the use of guide than providing redundant information already available with respect to the improvements between the HoloLens 1 and the HoloLens 2.

The biggest concern with the data and arguments such as "the errors obtained in this study are much lower than the ones presented in [1]" is that the manuscript does not present how many samples were collected, nor a study design that supports the stats done. 

The statement "several analyses in experimental scenarios demonstrating the advantages of Microsoft HoloLens 2 over Microsoft HoloLens 1 regarding AR projection accuracy and ergonomics." is inaccurate, the manuscript does not present several analyses, it only uses two scenarios and does not demonstrates, it corroborates advantages of the HoloLens 2 that can be obtained by looking at the differences in technical specifications. Ergonomics was not studied and the survey was not standardized nor connected to the use cases.  

Author Response

Dear Reviewer,

We want to thank you for your insightful comments, questions, and your interest in our work. The replies to your reviews are in the attached document.

We appreciate all your suggestions, which encourage us to keep researching in this field.

Round 2

Reviewer 1 Report

Authors improved the manuscript very well so I recommend this article to be published.